# Structural Analysis of Environmental Literacy of Urban Residents in China—Based on the Questionnaire Survey of Qingdao Residents

Huawen Tian [1] and Shanshan Chen [2,*]

1    College of Humanities and Law, Shandong University of Science and Technology, Qingdao 266590, China;
     skd992912@sdust.edu.cn
2    School of Foreign Languages, Suzhou University, Suzhou 234000, China
*    Correspondence: yuntimanbu@ahszu.edu.cn

**Abstract:** Residents' environmental literacy is the basis of sustainable development. The structure of environmental literacy and the interaction among its elements is a very important topic, which has been rarely studied. By using the literature analysis method to analyze the existing research of scholars, it can be concluded that the connotation of environmental literacy is composed of environmental knowledge, environmental awareness, and environmental behavior. The environmental literacy and structure of Qingdao residents can be measured by questionnaire survey. It has been found through a more in-depth structural analysis that environmental knowledge and environmental awareness will influence but cannot decide environmental behavior all by themselves; high-level environmental knowledge and high environmental awareness do not necessarily lead to a high-level environmental behavior but low-level environmental knowledge and low environmental awareness almost inevitably lead to a low-level environmental behavior. This finding is of great value for the government to formulate environmental policies scientifically, carry out environmental education more effectively, comprehensively improve residents' environmental literacy and improve their environmental behavior.

**Keywords:** environmental literacy; values; environmental education; environmental behavior

## 1. Introduction

In the first year of the 14th Five-Year Plan period (The 14th Five-Year Plan is China's 14th Five-Year Plan. The Five-Year Plan is an important part of China's national economic plan. It mainly sets out plans for major national construction projects, the distribution of productive forces and the important proportion of the national economy, and sets out goals and directions for the long-term development of the national economy. China formulated its first Five-Year Plan in 1953 and its 14th in 2021), China officially proposed the strategic goal of "carbon peaking and carbon neutrality". The "dual carbon" strategy requires the shared participation of all our people and will naturally raise the requirements of people's environmental literacy. So, in the context of the "dual carbon" strategy, the status of the environmental literacy of Chinese residents becomes particularly attractive.

Research on environmental literacy in the West is obviously older than that of China. As early as 1968, Roth, an American scholar, first proposed the concept of "environmental literacy". Early researchers, including Roth, have paid more attention to the definition and connotation interpretation of environmental literacy. This paper uses the research method of questionnaire survey, taking Qingdao residents as the object of investigation, trying to grasp the basic situation of residents' environmental literacy and its composition through questionnaire surveys. Qingdao is one of the most economically developed cities in China. It is located on the shore of the Yellow Sea and enjoys a beautiful environment. It is an important tourist city in northern China with a permanent urban population of

about 6 million. Qingdao has always attached great importance to urban environmental construction and environmental quality education. From as early as the 1980s, the publicity and education work on environmental protection began. In recent years, environmental education has been included in primary and junior high school education. It can be said that the residents of Qingdao have accumulated certain environmental knowledge and consciousness, and have higher environmental quality. The above points are common in Chinese cities, especially in big cities. Therefore, the selection of Qingdao residents as the research object is representative to a certain extent, and the research conclusions can basically match most other cities' environmental literacy [1–3]. As environmental literacy becomes gradually familiar to the public, the scope of research in this field is expanding. For the research, environmental literacy education and environmental literacy measurement are the two topics that the greatest attention is paid to. The former focuses on the education objectives, [4] education programs, [5] and the education implementation paths with libraries, governments, and various schools as the main participants [6,7]. The latter highlights the development of various measuring tools and actual measurements for specific groups [8–10]. Since entering the new century, the focus of research has shifted toward more micro themes, such as environmental awareness and environmental behavior. In particular, with increasing research on the influencing factors of environmental behavior in recent years, influencing factors such as internal personality psychology, external codes of ethics, and government governance, have been repeatedly mentioned by scholars [11,12]. Both environmental awareness and environmental behavior have been regarded by most scholars as parts of environmental literacy. Thus, recent research has seen a new focus, i.e., the structure of environmental literacy. Scholars have realized the complex composition of environmental literacy and started to explore its internal composition, such as the complex linkage between environmental cognition and environmental behavior, environmental awareness and environmental behavior. According to different research topics, the existing research can be subdivided into two categories:

(1) Environmental knowledge and environmental behavior. Scholars carried out studies from different perspectives and targeted different groups. For example, Xue Caixia and Li Hua found that environmental knowledge had a positive regulating effect on pro-environmental behavior when they studied the environmental behavior of tea farmers. Li Wenming et al. [13] carried out a study on tourists in a scenic spot and found that environmental knowledge in different environmental knowledge groups had different regulatory effects on environmental behavior; on high knowledge groups it had negative regulatory effects [14]. Liu Jiankun and Zhang Yunliang found that environmental knowledge played an important mediating role in the study of internet netizens' pro-environment behaviors [15];

(2) Environmental awareness and environmental behavior. Scholars have also carried out a certain amount of research on this problem, but put forward contradictory views. For example, Teng Yuhua, Chen Danni, and Rao Hua found that farmers' energy saving emotion and ecological values were the main factors affecting energy saving behavior [16]. Lv Weixia and Wang Chaojie proposed that environmental awareness plays an important intermediary role in the mobilization of garbage classification when studying garbage classification [17]. Ouyang Bin et al. also found that environmental awareness has a significant positive effect on Chinese residents' environmental behavior [18]. Of course, some scholars hold the opposite view. For example, He Qi believes that the individual environmental protection attitude does not translate into effective environmental behavior [19]. The research group of building energy conservation in Tsinghua University even proposed that the public's energy consumption has nothing to do with the awareness of energy conservation, but it is more influenced by their own socioeconomic status [20].

It can be seen that although the current literature has addressed the "structure of environmental literacy", the research is still basically centered on environmental behavior and surrounds the factors or mechanisms that affect environmental behavior. There are few or no studies focusing on the internal structure of environmental literacy and the logical

relations among its components. In fact, the current academic community has not formed a clear and unified understanding of the structure of environmental literacy, which is still a relatively blank field. Based on this, this paper, on the basis of the scientific measurement of urban residents' environmental literacy, furthers the in-depth analysis of its internal structure in order to discover the relationship between various elements of environmental literacy and the deep logic of mutual influence. We believe that this study will enrich the research on the "structure of environmental literacy", and also give people a more three-dimensional and in-depth understanding of environmental literacy.

## 2. Data Sources and Methodology

This paper uses the research method of questionnaire survey, taking Qingdao residents as the object of investigation, trying to grasp the basic situation of residents' environmental literacy and its composition through a questionnaire survey. Qingdao is one of the most economically developed cities in China. It is located on the shore of the Yellow Sea and enjoys a beautiful environment. It is an important tourist city in northern China with a permanent urban population of about 6 million. Qingdao has always attached great importance to urban environmental construction and environmental quality education. From as early as the 1980s, the publicity and education work on environmental protection began. In recent years, environmental education has been included in primary and junior high school education. It can be said that the residents of Qingdao have accumulated certain environmental knowledge and consciousness, and have higher environmental quality. The above points are common in Chinese cities, especially in big cities. Therefore, the selection of Qingdao residents as the research object is representative to a certain extent, and the research conclusions can basically match most other cities. The idea and process of the survey are as follows.

### 2.1. Indicator System Design

This paper examines the structure of the environmental literacy of residents, i.e., the parts of environmental literacy and their relationship. Thus, the first step of this study is to clearly define the connotation of environmental literacy and design an indicator system for measurement accordingly. Chinese and international scholars vary in their understanding of the connotation of environmental literacy. The following are some representative viewpoints, which are mainly from those scholars with high reputation or whose articles have high citations (see Table 1).

It can be seen from the above review that although scholars differ in their understanding of environmental literacy, these understandings share some obvious commonalities. The keywords above can be classified into the following categories through comparative analysis and classification. First, the cognitive category. This category includes cognition, knowledge, skills, perception, sensitivity, and ability, among others, and indicates "having certain environmental knowledge". Second, the category of values. This category encompasses attitude, emotion, evaluation, awareness, etc., and can roughly be referred to as "having high environmental awareness". Third, the category of action. This category includes participation and behavior and can be understood as "adopting positive environmental behavior". The above three categories constitute the three elements of environmental literacy. It is obvious that the first two are implicit elements while the latter are explicit ones. In this study, the elements of environmental literacy are identified and analyzed structurally (the so-called structural analysis is to explore the three elements and their relationship) to provide a basis for the development of the evaluation indicator system or the primary indicators.

**Table 1.** An overview of the academic community's understanding of environmental literacy.

| Scholar | Connotation of Environmental Literacy | Keywords |
|---|---|---|
| Roth [1] | Individuals have the desire and ability to make environmentally responsible decisions and take action to strike a balance between life quality and environmental quality. | Desire, ability, behavior |
| Hungerford et al. [2] | The connotation of environmental literacy is divided into three parts, namely, cognitive knowledge, cognitive process, and cognitive affection. | Cognition, knowledge, affection |
| Sia et al. [3] | It includes three elements, namely environmental knowledge, environmental awareness, and environmental behavior. | Knowledge, awareness, behavior |
| Marcinkowski T.J. [21] | 1. Perception of and sensitivity to the environment; 2. The attitude of respecting the natural environment; 3. Knowing how the natural system operates; 4. Understanding various local, regional, national, international, and global environment-related issues; 5. The ability to use first- or second-hand information sources to analyze, synthesize, and evaluate information on environmental issues, and evaluate these issues according to facts or personal values; 6. Actively and responsibly solving environmental issues through full devotion; 7. Acquiring the strategic knowledge to remedy environmental issues; 8. Having the skills to develop and implement relevant strategies and formulate plans to remedy environmental issues; 9. Active participation in the work of all classes to address environmental issues. | Perception, sensitivity, attitude, knowledge, cognition, evaluation, action, skills, participation |
| UNESCO [22] | 1. Perception of and sensitivity to the overall environment; 2. Understanding and having experience of environmental issues; 3. Having values and emotions for the environment; 4. Having the skills to identify and solve environmental issues; 5. Participation in addressing environmental issues that face all classes. | Perception, sensitivity, knowledge, emotion, skills, participation |
| ZENG Shaopeng [23] | Environmental literacy is the sum of knowledge, awareness, and behavior about humans' living environment acquired and formed through acquired learning. | Knowledge, awareness, behavior |
| CHEN Dequan, LOU Cheng Wu [9] | Environmental literacy is comprehensive literacy that is gradually learned and accumulated by people throughout life to ease the relationship between humans and the environment and human behavior toward the environment. | Comprehensive literacy |
| WANG Xaomei [24] | Environmental literacy is a comprehensive system that includes environmental knowledge, environmental emotion, environmental awareness, and environmental behavior. | Knowledge, emotion, awareness, behavior |

The three primary indicators are decomposed, thus the secondary indicators are obtained. The decomposition process is described as follows. First of all, in terms of environmental knowledge, the requirements for knowledge, such as perception, sensitivity, and other keywords, are comparatively low, and the knowledge only needs to be known, but the requirements for knowledge about skills and abilities are relatively high, and the knowledge needs to be entirely mastered. Therefore, environmental knowledge can be divided into two secondary indicators, namely, "shallow-level knowledge" and "deep-level knowledge". Secondly, in terms of environmental awareness, the three key words of attitude, emotion, and evaluation can represent the triple expression of values, and can be specifically expressed as the "attitude toward environmental issues", "emotion about nature" and "evaluation of environmental issues", respectively. These keywords are the three secondary indicators under this primary indicator. Finally, in terms of environmental behavior, participation represents "what has been done", which should be distinguished from "how to do". Both of them are also two secondary indicators in this category.

The tertiary indicators are specific environmental issues that can reflect or support the secondary indicators. Due to the existence of so many issues to choose from, only the five most representative issues are selected in this paper in terms of effectiveness and operability. There are three selection criteria for these issues. First, the issues must be closely related to the daily life of residents. Second, residents have full freedom of choice on these issues. Third, residents' choices may influence the effect of environmental governance. After repeated comparison and screening, the following five areas of issues were finally selected: domestic waste treatment, disposal of disposable dry batteries, environmental supervision and complaints, plastic and white pollution, automobile exhaust, and air pollution. Since it is necessary to quantitatively describe the residents' environmental literacy, each measurement indicator should be scored as well. Before assigning scores, the primary and secondary indicators should be assigned with weights. In this paper, the weight of each indicator was determined following experts' advice. A letter of inquiry was distributed to 12 experts and the responses were averaged. The value was fed back to all the experts for a second time to solicit opinions, and no objection was found. Therefore, the weight of each indicator was determined, and each three-level indicator was assigned points according to the percentage system principle, as shown in Table 2.

**Table 2.** Environmental literacy evaluation indicator system.

| Primary Indicator (Weight) | Secondary Indicator (Weight) | Tertiary Indicator | Score |
|---|---|---|---|
| Environmental literacy | Environmental knowledge (0.287) | Shallow-level knowledge (0.084) | Whether it is understood that different wastes need to be separately arranged | 1.68 |
| | | Whether the harm of waste batteries is understood | 1.68 |
| | | Whether we have the obligation to report environmental damage | 1.68 |
| | | … … … | 1.68 |
| | Deep-level knowledge (0.203) (0.203) | Recyclable waste | 4.06 |
| | | … … … | 4.06 |
| | Environmental awareness (0.296) | Attitude toward environmental issues (0.101) | Whether waste classification is the duty of citizens | 2.02 |
| | | … … … | 2.02 |
| | | Emotion about nature (0.097) | Feeling heartbroken when seeing "the hurt domestic waste causes animals" | 1.94 |
| | | … … … | 1.94 |
| | | Evaluation of environmental issues (0.102) | Opinions on wastes encircling cities | 2.02 |
| | | … … … | 2.02 |
| | Environmental behavior (0.417) | Environmental activities participated in (0.212) | Whether there was participation in activities related to waste classification, such as volunteer activities and collecting wastes in public places, etc. | 4.24 |
| | | … … … | 4.24 |
| | | | … … … | 4.10 |
| | | How to cope with environmental issues (0.205) | Almost always driving by yourself or your family when traveling for a short distance | 4.10 |
| | | | Whether waste bags are often used when shopping | 4.10 |

Note: limited by space, some tertiary indicators are omitted.

As far as the composition of environmental literacy is concerned, environmental behavior is explicit and has a more direct impact on the results; environmental knowledge and environmental awareness are implicit and have an indirect impact on the results. Thus, for environmental literacy, environmental behavior is more important than the last two. Among the two secondary indicators of environmental knowledge, the impact of "deep-level knowledge" is significantly greater than that of "shallow-level knowledge"; it is difficult to distinguish the importance of the three secondary indicators of environmental awareness and the two secondary indicators of environmental behavior. The two secondary indicators of environmental behavior, "environmental activities participated in" and "how to deal with environmental issues", are the actual environmental behavior, with roughly equivalent importance. It can be known through the above analysis that the weight distribution of each indicator in the table above is reasonable.

## 2.2. Survey Process and Reliability and Validity Analysis

In the urban area of Qingdao city, questionnaires were randomly distributed to pedestrians in the five most densely populated commercial districts or scenic spots, including Hong Kong Middle Road, Badaguan Road, Taitung Road, 4 May Square, and Huangdao Changjiang Road. The survey was conducted from April to May 2021, and the questionnaires were distributed during the Qingming Festival (3–5 April), Labor Day (1–3 May), and the Dragon Boat Festival (12–14 May). A total of 600 questionnaires were distributed within the urban area of Qingdao, 575 of which were recovered and 547 were valid, with a recovery rate of 95% and a validity rate of 91%. The recovery rate and validity rate of the questionnaires were high, which met the basic requirements of the survey. SPSS25.0 was used to process the questionnaires and test their reliability and validity. The Cronbach's alpha coefficient used for reliability analysis was 0.662, which met the requirement for medium reliability ($0.35 < \alpha < 0.70$). The validity analysis result showed that the KMO value was 0.768 and the Bartlett $p$-value was 0, which meets the analysis requirements.

## 3. Survey Results and Analysis

According to the survey results, the overall environmental literacy of Qingdao residents was at an average level, with an average score of 49.67 points only (out of 100 points). The highest score was 90 points and the lowest score was only 7 points, with the scores of most respondents lying between 30 and 60. The structural status and analysis of environmental literacy are as follows.

### 3.1. Structure of Environmental Literacy

The connotation of environmental literacy is divided into three parts, namely environmental knowledge, environmental awareness, and environmental behavior, which are also the three primary indicators of this study. The three parts form a unity but they maintain their independence. The three primary indicators are given different weights according to the degree of their importance. The scoring rate (the proportion of the actual score in the total score) can be used to measure their actual situation and further show the structure of environmental literacy. See Table 3 for the actual score, scoring rate, and the proportion of the actual score for the three primary indicators.

**Table 3.** Basic situation of the environmental literacy structure.

| Primary Indicator | Average Score | Scoring Rate | Weight/Proportion of Actual Score |
|---|---|---|---|
| Environmental knowledge | 13.75 | 45.83% | 28.7%/30.0% |
| Environmental awareness | 20.03 | 66.77% | 29.6%/43.8% |
| Environmental behavior | 15.79 | 39.98% | 41.7%/26.2% |
| Environmental literacy | 49.67 | 49.67% | 100.0%/100.0% |

It can be seen from the above table that among the three primary indicators, environmental awareness has the highest scoring rate, while environmental behavior has the lowest scoring rate, with the difference between the two being 26.79 percentage points, which can be said to be a huge gap. Moreover, there is a certain gap between the scoring rate of environmental behavior and that of environmental knowledge, with a difference of 5.85 percentage points. In terms of the proportion of scores, the actual score of environmental awareness has the highest proportion, 43.8%, far higher than 30%. On the contrary, the actual score of environmental behavior has a share of only 26.2%, far lower than 41.7%. Through the comparison of these two groups of data, it can be seen that the three parts of environmental literacy, namely environmental knowledge, environmental awareness, and environmental behavior, show a more significant disharmony. The overall level of environmental awareness is higher than that of environmental knowledge, and the overall level of environmental knowledge is higher than that of environmental behavior.

### 3.2. Interactive Analysis of Various Parts of Environmental Literacy

The above analysis reflects the overall situation of the environmental literacy structure or the gap among environmental knowledge, environmental awareness, and environmental behavior. However, is there any relationship between the three, and how do they influence each other? A further interactive analysis is, thus, required to answer these questions. For the convenience of analysis, we divided the overall level of each primary indicator into three levels: high, medium, and low. Among them, any of the indicators were considered to be at a high level if their scoring rates were higher than or equal to 70%; at a medium level if their scoring rates was lower than 70% but not lower than 40%, and at a low level if their scoring rates were lower than 40%. In this way, we could calculate the number and percentage of cases for each indicator at different levels and further carry out interactive an analysis of these indicators. The specific analysis results are as follows.

#### 3.2.1. Interactive Analysis of Environmental Knowledge and Environmental Awareness

First, the interactive analysis of environmental knowledge and environmental awareness was conducted, and the analysis results are shown in Table 4.

**Table 4.** Interactive classification of environmental knowledge and environmental awareness (person).

| Environmental Knowledge | Environmental Awareness | | | Total |
|---|---|---|---|---|
| | **High Awareness** | **Medium Awareness** | **Low Awareness** | |
| High-level knowledge | 28 | 5 | 0 | 33 |
| Medium-level knowledge | 248 | 124 | 0 | 372 |
| Low-level knowledge | 64 | 62 | 16 | 142 |
| Total | 340 | 191 | 16 | 547 |

As can be seen in the table above, among all the cases, the numbers for the cases with medium-level knowledge and high awareness are the largest, 248 and 340, respectively, accounting for 45.34% and 62.16%, while the number of cases with high-level knowledge is 33, accounting for only 6.03%, which further demonstrates that the environmental awareness of the surveyed population is generally ahead of their environmental knowledge. It is known by observing the distribution of cases with high- and low-level knowledge; among cases with high-level knowledge, the numbers for the cases with high, medium, and low awareness are 25, 5, and 0, respectively. It can be seen that high-level knowledge can help develop high awareness and effectively avoid low awareness; among the cases with low-level knowledge, the numbers for the cases with high, medium, and low awareness are 64, 62 and 16, respectively. It is clear that low-level knowledge does not affect the development of high awareness. Then, through observation of the distribution of cases with high and low awareness, among the high awareness cases, those with high-, medium-, and low-level knowledge are 28, 248 and 64, respectively. It is clear that high-level knowledge

does not necessarily lead to high awareness and low-level knowledge is the precondition of low awareness. According to the above analysis, the environmental awareness of the surveyed population is obviously ahead of their environmental knowledge, and high-level environmental knowledge cannot necessarily help establish high environmental awareness, but can effectively avoid the establishment of low environmental awareness; low-level environmental knowledge is a necessary condition for low environmental awareness.

### 3.2.2. Interactive Analysis of Environmental Knowledge and Environmental Behavior

The results of the interactive analysis of environmental knowledge and environmental behavior of all the cases are as in Table 5.

**Table 5.** Interactive classification of environmental knowledge and environmental behavior (person).

| Environmental Knowledge | Environmental Behavior | | | Total |
|---|---|---|---|---|
| | **High-Level Behavior** | **Medium-Level Behavior** | **Low-Level Behavior** | |
| High-level knowledge | 10 | 12 | 11 | 33 |
| Medium-level knowledge | 46 | 153 | 173 | 372 |
| Low-level knowledge | 5 | 43 | 94 | 142 |
| Total | 61 | 208 | 278 | 547 |

It can be seen from the statistical results in the table above that there are relatively few cases with high-level knowledge and high-level behavior in all cases; 32 and 61, respectively, accounting for 5.57% and 10.61%; the numbers for the cases with medium-level knowledge and medium-level behavior are the largest, 372 and 210, accounting for 64.70% and 36.52%, respectively, which is consistent with the point distribution of environmental literacy scores. Further analysis showed that high-, medium-, and low-level behaviors in all cases with high-level environmental knowledge are 10, 12 and 11, respectively, which shows that high-level knowledge produces no significant impact on environmental behavior; among all the cases with low-level knowledge, those with high-, medium-, and low-level behaviors are 5, 45, and 94, respectively. It can, thus, be seen that low-level knowledge makes a significant impact on environmental behavior, and can hardly help generate high-level behavior. Through further analysis, it can be found that among all the cases with high-level behavior, the numbers for the cases with high-, medium-, and low-level knowledge is 10, 46 and 5, respectively. It is evident that high-level behavior requires certain knowledge as its basis; among all low-level behavior cases, the numbers for the cases with high-, middle-, and low-level knowledge are 11, 173, and 94, respectively. It shows that low-level knowledge may affect environmental behavior. Based on the above analysis, the following conclusions can be drawn. The overall degree of matching between environmental awareness and environmental behavior is not high; high-level environmental knowledge may not cause high-level environmental behavior, but low-level environmental knowledge may cause low-level environmental behavior; environmental behavior requires certain environmental knowledge as its basis, i.e., environmental knowledge is a necessary condition for environmental behavior.

### 3.2.3. Interactive Analysis of Environmental Awareness and Environmental Behavior

The results of interactive analysis on environmental awareness and environmental behavior for all the cases are as in Table 6:

It can be seen from the table above that the total number of cases with high awareness is 340, accounting for 62.16%, significantly more than the number of cases with high-level knowledge and high-level behavior and there is a small number of cases with low awareness, only 16, accounting for 2.93%. This result is in line with the high scoring rate and the proportion of high scores of environmental awareness in the above statistics. Further analysis was made on the distribution of high and low-awareness cases. It is shown that the number of high, medium and low awareness cases is 51, 140 and 149, respectively. It is

evident that high awareness has no significant impact on environmental behavior and will not necessarily lead to high-level behavior; the numbers for the cases with high-level behavior, medium-level behavior and low-level behavior among the cases with low awareness are 0, 0 and 16, respectively, all of which are low-level behavior. Thus, it can be determined that low awareness will inevitably cause low-level behavior. The distribution of high-level and low-level behavior cases was then analyzed. Among high-level behavior cases, the numbers for the cases with high awareness, medium awareness and low awareness are 51, 10, and 0, respectively. It can be seen that high-level behavior requires high awareness as support; among low-level behavior cases, the cases with high, medium and low awareness are 149, 113 and 16, respectively. It can be seen that high awareness may also cause low behavior. However, considering the overall high environmental awareness, the proportion of the low awareness cases in all the cases with low-level behavior is 5.76%, higher than the proportion of low awareness cases in all cases (2.93%), indicating that low awareness is an important contributor to low-level behavior. To sum up, it can be found that environmental awareness and environmental behavior do not match each other, and environmental awareness is obviously ahead of environmental behavior; high environmental awareness does not necessarily lead to high-level environmental behavior, but low environmental awareness will inevitably lead to low environmental behavior; high-level environmental behavior requires high environmental awareness as support, while low environmental awareness is an important influencing factor of low-level environmental behavior.

**Table 6.** Interactive classification of environmental awareness and environmental behavior (person).

| Environmental Awareness | Environmental Behavior | | | Total |
| --- | --- | --- | --- | --- |
| | High-Level Behavior | Medium-Level Behavior | Low-Level Behavior | |
| High awareness | 51 | 140 | 149 | 340 |
| Medium awareness | 10 | 68 | 113 | 191 |
| Low awareness | 0 | 0 | 16 | 16 |
| Total | 61 | 208 | 278 | 547 |

3.2.4. Interactive Analysis of Environmental Knowledge, Environmental Awareness, and Environmental Behavior

The three primary indicators of environmental knowledge, environmental awareness, and environmental behavior are analyzed interactively, and the results are shown as in Table 7.

It can be seen from the data in the table above that the total number of cases with a relatively high level of environmental knowledge and environmental awareness, both of which are at the medium level or above, is 405, with a share of 73.84%, 24.47 percentage points higher than those with high-level environmental behavior; the numbers for the cases with high-, medium-, and low-level environmental behaviors among the cases with high-level environmental knowledge and high environmental awareness are 56, 165 and 184, respectively. It can, thus, be concluded that environmental knowledge and environmental awareness are generally ahead of environmental behaviors, but cannot determine environmental behaviors. Among the cases with low-level environmental knowledge and low awareness, at a low rather than high level, the numbers for the cases with high-, medium-, and low-level behaviors are 1, 25 and 54, respectively. It is clear that low-level environmental knowledge and low awareness will produce an impact on low environmental behavior, and it is difficult for them to cause high-level environmental behavior. The above viewpoints are consistent with those of the previous analysis.

There is a question mark over this conclusion, which may require further explanation. The study found that residents' environmental awareness was relatively ideal, with a score rate of 66.77%, which was significantly ahead of environmental knowledge and environmental behavior, especially in sharp contrast with environmental behavior. Is this a reasonable phenomenon? How does high environmental awareness develop? As

mentioned in this paper, the residents have a relatively great environmental awareness, with a scoring rate of 66.77%, obviously ahead of environmental knowledge and environmental behavior, forming a strong contrast with environmental behavior in particular. However, does this phenomenon make sense? In fact, such a phenomenon has been found in many existing studies. For example, in the field of waste classification, a survey conducted in Shanghai found that 90% of the residents supported waste classification as early as 2011, but the accuracy of waste classification in pilot areas was lower than 20% [25]. By 2019, when the awareness of residents in Beijing toward the four categories exceeded 80% and their support rate exceeded 90%, the correct classification rate was lower than 20%. It can be seen that urban residents in China generally have high environmental awareness and low-level environmental behavior, which can be explained by the following two aspects. First, the limitations of the survey method. Currently, the questionnaire survey method is used in most of the existing studies, which is the same as that in this paper. As is known to all, although this method has many advantages, it also has some defects that are difficult to overcome, such as the lack of scientific attitude from respondents, and the insufficient depth of the survey. Particularly, the limitations of the questionnaire are most obvious when a survey is made on issues such as thoughts, motivations, concepts, values, etc. For the survey on environmental awareness, the questions designed by the surveyors are mostly "what attitude they hold toward a certain environmental issue", which is undoubtedly related to self-cognition. The "D-K effect" in psychology has revealed that people's self-awareness is often biased, and most people will think highly of themselves. Thus, when asked such questions on paper, there will be a deviation between the respondents' answers and the truth. The characteristics of the questionnaire determine that surveyors can only collect the results of the respondents' answers, and cannot go deeper into their real thoughts. Therefore, the conclusions drawn based on the survey data are likely to be better than the actual situation. Given this fact, more survey means, such as interviews and observation, should be introduced in subsequent studies to obtain more vivid data. Second, China has not attached importance to the construction of ecological civilization for a long time, and the improvement of environmental literacy is a slow process, in which the improvement of environmental awareness is usually ahead of that of environmental behavior. The improvement of the public's environmental awareness (such as their understanding of some environmental issues) demonstrate good results by means of publicity within a relatively short time, but it seems that environmental behavior is more difficult to improve. That is because the improvement of environmental behavior may not only change the long-term living habits of residents but also add to the cost of living. Therefore, in addition to basic publicity means, we should also take more binding means to promote people's environmental behavior, such as economic regulation, information disclosure, and even administrative coercion. This does not only mean higher administrative costs for the government, but also some uncertain social risks. Therefore, in reality, the government adopts a variety of publicity means that feature low cost, low risk, and easy operation, while the use of more binding economic means and coercive means will require great caution. In this way, the final result is that the environmental awareness we see is far ahead of environmental behavior.

**Table 7.** Interactive classification of environmental knowledge, environmental awareness, and environmental behavior (person).

| Environmental Knowledge/Environmental Awareness | Environmental Behavior | | | Total |
| --- | --- | --- | --- | --- |
| | High-Level Behavior | Medium-Level Behavior | Low-Level Behavior | |
| High-level knowledge/high awareness | 10 | 9 | 9 | 28 |
| High-level knowledge/medium awareness | 0 | 3 | 2 | 5 |
| Medium-level knowledge/high awareness | 37 | 111 | 100 | 248 |

**Table 7.** *Cont.*

| Environmental Knowledge/Environmental Awareness | Environmental Behavior | | | Total |
| --- | --- | --- | --- | --- |
| | **High-Level Behavior** | **Medium-Level Behavior** | **Low-Level Behavior** | |
| Medium-level knowledge/medium awareness | 9 | 42 | 73 | 124 |
| High-level knowledge/low awareness | 0 | 0 | 0 | 0 |
| Medium-level knowledge/low awareness | 0 | 0 | 0 | 0 |
| Low-level knowledge/high awareness | 4 | 20 | 40 | 64 |
| Low-level knowledge/medium awareness | 1 | 23 | 38 | 62 |
| Low-level knowledge/low awareness | 0 | 0 | 16 | 16 |
| Total | 61 | 208 | 278 | 547 |

## 4. Conclusions and Prospect

### 4.1. Research Conclusions

It can be found through the comprehensive analysis of this paper that the overall environmental literacy of urban residents of Qingdao is not high. Going deeper into environmental literacy, it is also found that the urban residents of Qingdao have high environmental awareness, but a relative lack of environmental knowledge and even more insufficient environmental behavior. The three are not coordinated with each other. The three parts of environmental literacy, namely environmental knowledge, environmental awareness, and environmental behavior, interact with each other, which can be described as follows.

First, high-level environmental knowledge does not necessarily lead to high environmental awareness, nor does it necessarily lead to high environmental behavior. However, conversely, both high environmental awareness and high-level environmental behavior need certain environmental knowledge as their basis, and high-level environmental knowledge can avoid the establishment of low environmental awareness and low environmental behavior to some extent.

Second, low-level environmental knowledge is very likely to cause low-level environmental behavior.

Third, high environmental awareness does not necessarily lead to high environmental behavior, which needs a relatively high environmental awareness as its basis, though.

Fourth, low environmental awareness almost inevitably leads to low environmental behavior.

In conclusion, environmental knowledge and environmental awareness constitute two important influencing factors of environmental behavior. Although they cannot completely determine environmental behavior, they are indispensable prerequisites of the improvement of environmental behavior.

### 4.2. Prospect

The ultimate goal of environmental literacy research is to improve people's environmental behavior, but this is undoubtedly a very complicated problem, which can be roughly discussed from two aspects.

One is the internal factors that affect environmental behavior or the environmental knowledge and environmental awareness discussed in this paper. Although environmental knowledge and environmental awareness cannot completely determine environmental behavior, they constitute important influencing factors and preconditions for improving environmental behavior. Currently, despite the proper awareness of urban residents in China, they have relatively insufficient environmental knowledge, with a shortage of high-level environmental knowledge in particular. In this case, education is the top priority in promoting high-level environmental knowledge. First, school education. In school education, at various levels, it is necessary to truly attach importance to environmental education, optimize the curriculum system of environmental education, and increase the

share of environmental education in assessments and further education [26]. Second, social education. It is essential to widely publicize environmental-protection-related knowledge through various media, organize publicity activities on environmental knowledge together with communities and work units, add the social supply of environmental-type popular science books, and guide the integration of environmental concepts and knowledge in various literary and artistic works. The other is external factors. Environmental knowledge and environmental awareness are the only necessary conditions for environmental behavior. Certain "boosting" means, such as policy guidance, market guidance, cultural guidance, and necessary administrative coercion measures, are required to truly change people's environmental behavior [27,28]. Of course, this is a more complex problem that requires special research from the perspectives of psychology, economics, public management, public policy, and other disciplines.

This paper conducts a tentative study of environmental literacy in China and provides some discoveries, but not without shortcomings. For example, there are limitations to measuring environmental awareness with a questionnaire survey; no further research on environmental behavior is conducted. These issues need more in-depth research and more active exchanges from our peers in academic circles. It is hoped that this paper can start further discussions among our peers.

**Author Contributions:** Conceptualization, H.T.; Methodology, H.T.; Formal analysis, S.C.; Investigation, S.C.; Resources, S.C.; Writing—original draft, H.T.; Writing—review & editing, S.C.; Project administration, S.C. All authors have read and agreed to the published version of the manuscript.

**Funding:** The Social Science Planning Project in Qingdao (QDSKL2201142); the Innovation and Entrepreneurship Support Program for Returned Overseas Students in Anhui Province (2022LCX024); the Scientific Research Project (Philosophy and Social Science) of Anhui Higher Education Institution (2022AH051347); the Doctoral Research Start-up Fund Project of Suzhou University (2022BSK002).

**Institutional Review Board Statement:** Not applicable.

**Informed Consent Statement:** Not applicable.

**Data Availability Statement:** The data presented in this study are available from the authors upon request.

**Conflicts of Interest:** The authors declare no conflict of interest.

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
