# Peer review of "Structural Analysis of Environmental Literacy of Urban Residents in China—Based on the Questionnaire Survey of Qingdao Residents"

_sustainability, doi:10.3390/su15065552_

Round 1

Reviewer 1 Report

A growing body of literature has examined environmental aspects of sustainability. The submitted manuscript is in line with this trend. As was noted in the paper: “(...) this paper conducts a further internal structural analysis of the environmental literacy of urban residents by scientifically measuring its structure to identify the relationship among the elements of environmental literacy and the deep logic behind their mutual influence.” The subject is novel and very important, but unfortunately, the results fail to address the aim of the research. The conclusions do not support the results, which is a serious shortcoming. Urban residents of Qingdao (who were barely described) were questioned and it was treated as a main method of the research. Therefore, the Reviewer does not see the possibility of spreading the results to all the Chinese population or even ‘urban residents’ at all. Moreover, the results are too generalized. The studied population is also characterized by an extreme value of environmental literacy, e.g. very high or very low knowledge. Therefore, a deeper statistical approach is needed. Is it possible to describe the population and the results by Gaussian distribution or any other? Generalization is very important because it describes the average values, but the other values (extremely high or low) are also very important. Especially if the Authors tend to prepare the scientific basis for shaping government policies, which should also include minority of the population. Furthermore, it was not explained why urban residents of Qingdao were questioned. It must also be mentioned that some parts of the manuscript should be strengthened in scientific soundness and provide a clearer discussion of the investigation. The discussion should be supplemented by a comparison of the results with another research investigating a similar phenomenon. The Authors should place the study in a broader research context. Accordingly, the Reviewer also has the following suggestions for further revision:

1.    The questionnaire form is missing. The reviewer does not know what questions were asked. It should be added as supplementary material or as additional material only for the peer review process.

2.    The introduction section [1] is missing. The initial phase of the manuscript should be named the first section. Moreover, this section explained the research background and the purpose of the work; however, an explanation of the significance of this research is missing. It should be explained how this research will supplement the current body of literature.

3.    The 14th Five-Year Plan Period should be explained for non-Chinese readers (first line of the body of the manuscript).

4.    Table 1 shows "some representative viewpoints" of environmental literacy. On what methods were these viewpoints chosen and why are they representative?

5.    The keyword "process" is missing (should be added?) in the second line (Hungerford et al.) of Table 1. What is the impact of the cognitive process on environmental literacy?

6.    The reference 'Marcinkowski [16]' is misspelled in Table 1.

7.    The number and structure of experts should be explained. Furthermore, the average value is statistically weak (Table 2). The reviewer is not aware if the experts were consistent or had different opinions about the weight of indicators. If you tend to scientifically establish the evaluation system, your research should be precisely described and show as many statistical explanations as needed.

8.    The dots in Table 2 (tertiary indicator column) are not understandable.

9.    It is not understandable why the Discussion section is structured before the Conclusions section. The conclusions should also include elements of discussed results. Moreover, the conclusions were prepared for environmental knowledge, awareness, and behaviour, but not for environmental literacy at all. The comprehensive and holistic approach is missing.

10. Tables 4-6 could be supplemented by % values.

11. The improvement of environmental behaviour explained in the Discussion section was not the aim of this paper. This aspect could be taken into account in another paper or even a monograph.

I believe that rejecting this manuscript at the moment and overall revision will help the Authors with further work and provide scientifically grounded research on environmental literacy.

Reviewer 2 Report

The title could be simplified.

The abstract is not clear. Please check the significant elements of the abstract and explain all of them. The problem statement, research method, result and the study's contribution are not clearly defined.

The authors need to define environmental literacy in the introduction.

You should start with the introduction as 1.0.

In the introduction, the citations are confusing, why OUYANG Bin and HE Qi were written in the capital as this can just the put as citations?

All the tables are not referred to at all in the text. 

'1. Data sources and Methodology' part is mixed between methodology and literature/theory.  

What is the population and sampling of the study? Which method has been used?

Figure 1, the quality of the chart is poor.

Please do a comprehensive and critical literature analysis on the subject based on the current research papers, and relate it to the problem statement.

The reference too old (more than 5 years) should be strengthened with the current publications.

The organisation of the paper could be improved, end the manuscript with a conclusion and references. The Discussion should come earlier,

Overall, the paper presented a descriptive analysis of a survey. There is nothing new as the study just reported on the data analysis briefly. Is there any analysis that could be further added to the manuscript so will be more new findings that add value to the paper? 

Reviewer 3 Report

Here are the specific comments:

1. The bottom of page 3:

In the three levels of indicators, there are more specific environmental issues to choose from. The reasons why the two aspects of effectiveness and operability (and not others) were chosen to start the discussion are not stated in the text. The author is requested to pay attention to the smoothness of the logical connections.

2. Table 2:

Regarding Table 2, the interpretation of the weighting coefficients is reasonable, but there are no direct research findings to base the weights on. Please explain the reason or basis for taking the current weights for the main indicators and point out the corresponding literature or research findings.

3. Figure 1:

Regarding Figure 1, please standardize the diagram. The units of the data of the coordinate axes need to be clearly indicated. In addition, please express the content of Figure 1 in mathematical language such as distribution functions or trends, rather than leaving it to the reader to analyze it twice. In addition, please improve the clarity of the graphs.

4. Section 1.2: Please indicate when the questionnaire was obtained to ensure the validity of the data.

5. Section 2.3: The interaction analysis in the results section of the paper has only a qualitative approach to the study, and no specific interaction analysis model is seen. If so, please add it and state exactly what mathematical model was used to achieve it.

Moreover, in section 2.3: Indirect conclusions please consider placing them in the discussion section rather than the results section.

6. Please streamline the conclusion.

7. The number of references is not sufficient, please enrich the theoretical basis of the article after extended reading and citation.

8. The research target of the article is Qingdao residents, but the conclusion and discussion section evolves directly to the national context, and the conclusion of the study has no direct conclusion about the national environmental literacy situation, perhaps it can be viewed as an outlook. If possible, please keep the wording of the scientific and technical paper accurate.

Reviewer 4 Report

This research examines the internal structural analysis of the environmental literacy of urban residents to identify the relationship among the elements of environmental literacy and the deep logic behind their mutual influence. The study focuses on environmental knowledge, environmental awareness and environmental behavior. The topic is both important and interesting. However, the following remarks must to be taken into account:

1)      The ‘Title’ is too long. I propose removing "based on a questionnaire survey of Qingdao residents."

2)      Information about the fund project and about the authors should be at the end, before the list of references.

3)      If you decided to remove "Qingdao residents" from the title, you need to add the context of the study in the abstract.

4)      On the keywords, add (;) after values.

5)      The citations (references in text) should be regular without a superscript effect.

6)      The structure of the manuscript should be corrected. The headline is missing from the first paragraph of the manuscript. This should be the ‘Introduction’. Accordingly, the numbering of the other sections should be corrected. The "Conclusion" section should be the last section.

7)      The Introduction and background should be more closely linked with the aim of the study and strengthened with the support of additional theoretical recent references that might be provided with specific regards to the index categories.

8)      How was the scoring system for the indicators decided? I believe a sample of 12 authoritative experts is not enough to determine the weight of the indicators.

9)      The items of the questionnaire are not defined. You need to define them clearly.

10)   There should be a section with the result discussion in relation to literature and existing bodies of knowledge / practices.

11)   I believe that the ‘Discussion’ section at the end of the manuscript should be renamed, edited, and corrected. The content explores recommendations for future work and further research. In addition, it includes the limitations of the current study. These can be elements of the conclusion.

12)   The list of references should be enhanced with updated and related references.

Reviewer 5 Report

Dear Authors,

Thank you for the opportunity to review your paper.

The language is appropriate.

The abstract is well written - short and shows the essence of the text.

The introduction is well written as it introduces the reader to the subject matter.

Conclusions are also well written.

Notes on the rest of the text:

-          1. Perception of and sensitivity to the environment; 2. The attitude of respecting the natural environment; 3. knowing how the natural system operates; 4. understanding various local, regional, national, international, and global.

The form of the writing style should be the same – All sentences should start with either lowercase or uppercase letters. Correct this shape throughout the text.

-          The Conclusion section should be written after the Discussion section.

Round 2

Reviewer 1 Report

Dear Authors,

thank you for your reply. Unfortunately, it was hard to recognize which answers correspond with my review report, because there are no answers for all of my suggestions. However, I decided to track changes in the revised manuscript, therefore, I have the following suggestions for further revision:

1.    The statement “I believe that this study…” (last sentence in section 1) is confusing while the paper is signed by two authors.

2.    Table 1 still shows "some representative viewpoints" of environmental literacy. The authors’ respond: “The representative views in Table 1 are mainly from those scholars with high reputation or whose articles have high citations.” should be included in the body of the manuscript.

3.    The reference 'Marcinkowski [16]' is still misspelled in Table 1.

4.    The dots in Table 2 (tertiary indicator column) are not understandable. I also do not support the authors’ respond. Please, explain it in the manuscript.

5.    First and second sentence of 4.1. section should be rewritten due to residents of Qingdao rather than residents of China. The conclusion should be rewritten in this manner.

6.    I do not see reference to the following statements: “Similar results have been obtained from relevant surveys in other cities. By 2019, when the awareness of residents in Beijing toward the four categories exceeded 80% and their support rate exceeded 90%, the correct classification rate was lower than 20%.”. Moreover, the “other cities” is only Beijing? Please, be specific.

Author Response

Thank you for your comments on my manuscript. We have carefully considered your suggestion and made revisions according to your comments which we highlighted in red. Revision notes are given as follows:

  1. "I" was changed to "We".
  2. It has been explained in the manuscript.
  3. The spelling mistakes have been corrected.
  4. Notes have been made in the manuscript.
  5. It was revised to Qingdao resident.
  6. Deleted: "Similar results have been obtained from relevant surveys in other cities."

Reviewer 2 Report

I suggest author to reply a rebuttal by preparing the comments and answers as i cannot trace back all the answers. the references are still old, no answers on samping etc. i even don't know either i am reviewer 1, 2 or 3..

Author Response

Thank you for your comments on my manuscript. We have carefully considered your suggestion and made revisions according to your comments which we highlighted in yellow or red. Revision notes are given as follows:

  1. The modified parts are: (1) The title and abstract have been revised according to your comments; (2) A supplementary introduction to the research object the residents of Qingdao and sampling procedure was added; (3) After careful consideration, Figure 1 was deleted; (4) New references are supplemented and replaced; (5) The structure was modified , especially the conclusion and discussion parts were revised to conclusion and implication.
  2. My explanation: (1) There is no self-definition of environmental literacy, because it is controversial. Some representative views (mainly on definitions) have been listed in Table 1, from which the internal composition of environmental literacy can be derived. (2) most of the existing studies focus on environmental literacy as a whole and pay little attention to its internal composition.

Reviewer 3 Report

The authors have carefully considered our suggestions and made specific revisions which we highlighted in yellow.

Author Response

Thank you for your comments on my manuscript. We have made some new modifications to the article which we highlighted in red, thanks again.

Reviewer 4 Report

The revised version did not consider important comments raised in round 1. Some gaps still found.

1)      There should be a section with the result discussion in relation to literature and existing bodies of knowledge / practices.

2)      The list of references should be enhanced with updated and related references.

Author Response

Thank you for your comments on my manuscript. We have made some new modifications to the article which we highlighted in red.We would like to explain your question as follows:

  1. The last paragraph of the first section is the result of the discussion of the existing literature and the relationship between this study and the existing literature.
  2. The reference literature has been updated, and there are no other new representative literature retrieved so far.

Round 3

Reviewer 1 Report

Thank you for the revision of your manuscript.

Reviewer 4 Report

The new version of the paper addressed the concerns raised in the first and second review.